# Modulation-free laser stabilization technique using integrated cavity-coupled Mach-Zehnder interferometer

Mohamad Hossein Idjadi ®[1] ✉, Kwangwoong Kim[1] & Nicolas K. Fontaine ®[1]

Stable lasers play a significant role in precision optical systems where an electro-optic laser frequency stabilization system, such as the Pound-Drever-Hall technique, measures laser frequency and actively stabilizes it by comparing it to a frequency reference. Despite their excellent performance, there has been a trade-off between complexity, scalability, and noise measurement sensitivity. Here, we propose and experimentally demonstrate a modulation-free laser stabilization method using an integrated cavity-coupled Mach-Zehnder interferometer as a frequency noise discriminator. The proposed architecture maintains the sensitivity of the Pound-Drever-Hall architecture without the need for any modulation. This significantly simplifies the architecture and makes miniaturization into an integrated photonic platform easier. The implemented chip suppresses the frequency noise of a semiconductor laser by 4 orders-of-magnitude using an on-chip silicon microresonator with a quality factor of $2.5 \times 10^6$. The implemented passive photonic chip occupies an area of 0.456 mm$^2$ and is integrated on AIM Photonics 100 nm silicon-on-insulator process.

Precise laser frequency control is a crucial requirement in various applications, including optical communication[1,2], optical atomic clocks[3,4], sensing[5,6], and photonic-assisted ultra-low phase noise microwave and mmWave synthesis[7–9], which makes stable and narrow linewidth lasers indispensable part of precision optical experiments. Researchers have explored various methods to suppress unwanted laser frequency noise using optical feedback[10,11], electro-optic feedback[12–14], and electro-optic feed-forward[15] techniques. In the optical feedback method, or self-injection locking (SIL), a small portion of the laser output is filtered and injected back into its cavity[16,17]. The SIL method is a promising way to suppress laser frequency noise across a large bandwidth. Although this method offers excellent short-term stability, frequency noise reduction over a large bandwidth, and significant linewidth suppression, it typically cannot guarantee long-term stability and requires active stabilization[18–20]. Moreover, designing the optical feedback usually mandates the co-design of a laser system and optical signals with an ultra-high quality factor (Q-factor) cavity. Electro-optic techniques, on the other hand, leverage state-of-the-art

and mature electronic devices and systems to precisely control the laser frequency and offer long-term laser frequency stabilization. This will push some of the challenges from the optical domain into the electrical one, where control and manipulation of the signals and systems are comparatively more manageable and cost-effective.

A key building block in electro-optic laser stabilization techniques is an optical frequency noise discriminator (OFND). An OFND measures the frequency fluctuations of a laser by comparing it to a frequency reference and generates an electronic signal corresponding to the frequency noise that can be further processed in the electrical domain. Different OFND configurations have been explored, such as the "squash" locking technique[21,22], the Pound–Drever–Hall (PDH) laser stabilization method[12], and the unbalanced Mach-Zehnder interferometer (MZI)[14,23]. The PDH method stands out as the most well-known precision laser instrumentation technique among the extensively utilized OFNDs[24–27]. Using the PDH technique, a sharp asymmetric error signal can be generated, which can then be utilized as a servo signal to stabilize the laser frequency. In PDH architecture, the

[1]Nokia Bell Labs, 600 Mountain Ave, Murray Hill, NJ 07974, USA. ✉e-mail: mohamad.idjadi@nokia-bell-labs.com

laser phase is modulated using a phase modulator, filtered by an optical frequency reference (e.g., Fabry–Pérot (FP) cavity) followed by a photodetector[28]. The photodetected signal is down-converted with the same local oscillator frequency to the baseband. This up/down conversion scheme makes PDH more robust against DC error signal offset, base-band noise, and other perturbations at low frequency, which enables a robust stable frequency lock. Despite excellent performance, the PDH scheme requires an electro-optic phase modulator and fast enough electronics for modulation and demodulation, which increases the power consumption and area for an integrated PDH chip[29,30]. It is worth mentioning that the properties of the electronics in the PDH loop, such as modulation speed, amplitude, and transimpedance gain, depend on the optical frequency reference characteristics, such as the cavity finesse. As a result, depending on the cavity finesse, the modulation speed may vary from MHz to GHz range. Moreover, as far as integration level is concerned, ultra-high Q-factor integrated cavities are usually implemented in platforms (e.g., silicon nitride) that lack efficient electro-optic phase modulation in C-band. This makes full integration of the PDH system in a single photonic chip challenging.

Alternatively, a passive-only unbalanced MZI can serve as an OFND where the two arms of the MZI are phase-locked at the quadrature point[31]. Although the unbalanced MZI has a simple architecture to implement on a chip, achieving high sensitivity frequency detection comparable to the PDH method requires either a large optical delay line or a substantial electronic gain that comes at the cost of chip area and overall system power consumption.

Here, we propose and experimentally demonstrate a modulation-free laser stabilization technique using a cavity-coupled MZI on a silicon photonic chip as an OFND. The proposed frequency noise discrimination technique utilizes a high Q-factor cavity coupled to an MZI, which breaks the trade-off between sensitivity, complexity, chip area, and power consumption. With a careful design of the on-chip high Q-factor cavity coupled to an MZI, about 4 orders-of-magnitude suppression in frequency noise of three commercially available distributed feedback (DFB) lasers is achieved. On-chip thermal tuners are

implemented for the potential trimming of fabrication-induced errors and for facilitating the broadband operation of the OFND. The proposed architecture combines the advantages of a passive-only structure, which offers simplicity, lower power consumption for error signal processing, and sensitive frequency noise detection similar to the widely utilized PDH technique. Moreover, the proposed architecture utilizes a balanced detection scheme that significantly suppresses the intensity noise of the laser by the common-mode rejection ratio of the balanced photodetectors[32].

The proof-of-concept integrated photonic chip is fabricated in AIM Photonics, a commercially available 100 nm silicon-on-insulator (SOI) process. The photonic chip occupies an area of 0.456 mm$^2$ and consumes only about 50 µW power for reverse biasing the balanced photodetector. The proposed architecture offers a promising solution for achieving sensitive, simple, and low-power laser frequency stabilization systems. The proposed method can be co-integrated with mature electronics[14,30], which sets the stage for the development of low-cost, scalable, and stable fully integrated lasers. It is worth emphasizing that our proposed laser stabilization technique is versatile and, similar to other laser frequency locking techniques (e.g., PDH), it can be applied across various platforms to achieve effective laser frequency noise suppression.

## Results
### The principle of operation
Figure 1a shows the block diagram of an electro-optic laser frequency noise reduction loop. As mentioned earlier, the key part of this loop is an OFND that senses the frequency fluctuations of the incoming laser signal, compares it with an optical frequency reference ($f_{ref}$), and generates an electronic signal whose amplitude is proportional to the intensity of frequency fluctuations. This error signal is then amplified in the electrical domain and fed back into the laser cavity to stabilize its frequency. The details of the closed-loop operation and the linearized block diagram of the control loop are discussed comprehensively in Supplementary Note 5. The OFND can mathematically be represented by an asymmetric transfer function where the small frequency

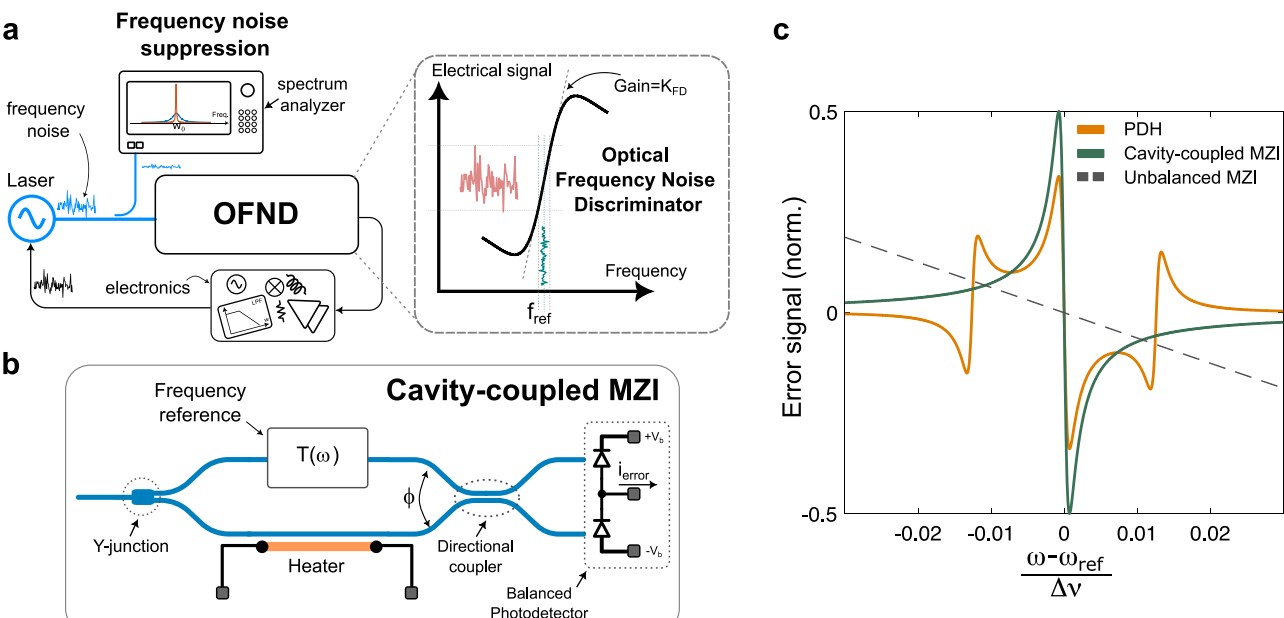

**Fig. 1 | The cavity-coupled MZI frequency discriminator. a** The block diagram of an electro-optic laser frequency stabilization using an optical frequency noise discriminator (OFND). The OFND response is asymmetric with respect to the frequency reference point, $f_{ref}$, and hence small frequency fluctuations translate into an electrical signal by the OFND gain. **b** The conceptual diagram of the proposed cavity-coupled MZI. $T(\omega)$ is the transfer function of a frequency reference (e.g., optical cavity) coupled into the top arm of the MZI. **c** Numerical analysis comparing the normalized error signal for the proposed cavity-coupled MZI, a conventional unbalanced MZI, and the PDH architecture. $\Delta f$ and $\Delta\nu$ are the laser offset frequency compared to $f_{ref}$ and the free-spectral range of the cavity.

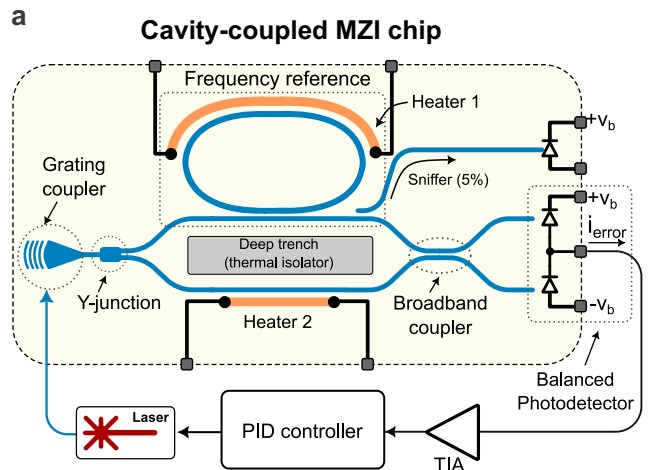

**Fig. 2 | The modulation-free laser stabilization scheme. a** The proposed cavity-coupled MZI is used as an OFND in a feedback loop for laser frequency noise suppression. A high Q-factor Euler microring resonator is utilized as a frequency reference. Half of the on-chip laser intensity is injected into the microring resonator coherently interfered with the bottom branch, and converted to an electrical signal using balanced Germanium photodetectors. A small portion of the ring resonator output is used to monitor its response. The generated error signal, $i_{error}$, is then amplified and fed into the PID controller. **b** The micro-photograph of the integrated photonic chip in AIM Photonics 100 nm silicon-on-insulator (SOI) process. The size of the photonic integrated circuit measures 0.95 mm × 0.48 mm. PID proportional-integral-derivative, TIA trans-impedance amplifier.

perturbation around $f_{ref}$ is amplified by a gain ($K_{FD}$) and converted into an electronic error signal. The slope of the transfer function also indicates the sensitivity of the OFND in measuring frequency noise.

Figure 1b shows the proposed cavity-coupled MZI as an OFND that maintains a simple passive structure without any need for fast optical phase modulation. In this method, the incoming laser intensity is split equally into two MZI branches using a broadband Y-junction. The top branch of the MZI is coupled into an optical frequency reference (e.g., an optical resonator or a cavity). The amplitude and phase of the electric field at the output of the cavity is affected by that of the frequency reference, $T(\omega)$. The bottom arm of the MZI is used to interfere with the frequency reference output electric field using a directional coupler. A balanced photodetector is used to photodetect the output of the MZI, and by subtracting the currents, the error signal ($i_{error}$) is generated. In other words, the proposed architecture is a coherent detector that uses the input laser signal (bottom arm of the MZI) to down-convert the optical signal at the output of the cavity. In this way the sharp asymmetric phase transition in the transfer function of the cavity at frequency of $f_{ref}$ will translate in a sharp electrical error signal that can be used to lock the laser in a feedback loop. The error signal at the output of the MZI can be written as (Supplementary Note 1)

$$i_{error}(\omega) = RP_0 |T(\omega)| \sin(\psi(\omega) - \phi), \tag{1}$$

where $|T(\omega)|, \psi(\omega), \phi, P_0$, and $R$ are the amplitude and phase of the optical reference transfer function at the frequency of $\omega$, phase difference between arms of the MZI controlled by a thermal phase shifter, intensity of the electric field at the input of the MZI, and the responsivity of the photodetectors, respectively. Figure 1c is the numerical analysis of the normalized error signal of the PDH, a conventional unbalanced MZI, and the proposed cavity-coupled MZI structure using Eq. (1). The details of the numerical comparison are presented in Supplementary Note 2. To assume a fixed area to implement the OFND, a 1 mm circumference ring resonator is used as the frequency reference in both the PDH and the cavity-coupled MZI. The length mismatch between the arms of the unbalanced MZI is also set to 1 mm. The waveguide loss is assumed 0.2 dB cm⁻¹. As shown in Fig. 1c, the error signal of the cavity-coupled MZI is significantly sharper than the conventional unbalanced MZI for the same given setting and, indeed, is comparable to that of the PDH, offering the same level of sensitivity but much simpler architecture.

To provide a fair comparison, although the proposed cavity-coupled MZI architecture offers a simple and sensitive frequency noise measurement technique, it demands careful design consideration. The PDH technique is less vulnerable to DC offset error and base-band noise. However, the frequency content of the proposed OFND is at the base band and more susceptible to unwanted low-frequency fluctuations, which demands careful noise and link-budget analysis (see Supplementary Note 4). Also, the slope of the OFND response depends on $\phi$ in Eq. (1) and has to be set properly. The random phase variation can be significantly mitigated by miniaturizing the system on a chip within a compact footprint. This results in a small temperature gradient across the integrated chip, thereby suppressing random phase difference fluctuations caused by temperature changes. For instance, in the closed-loop laser frequency locking experiments, we observed a stable and robust lock without active phase locking the MZI, even with the on-chip thermal phase shifters turned off. Supplementary Note 2 describes the comparison between different OFND architectures.

### Modulation-free laser frequency noise suppression system

Figure 2a shows the block diagram of the implemented modulation-free laser stabilization scheme using the implemented cavity-coupled MZI as an OFND. As illustrated in Fig. 2a, the laser output is coupled into the chip via a grating coupler and divided in half by a Y-junction. In the top branch, a high Q-factor microring resonator is used as an optical frequency reference which filters the amplitude and phase of the incoming light. The phase response of the ring resonator exhibits rapid and asymmetric change around its resonance. When combined with the amplitude response, this characteristic provides sufficient information to determine the offset between the laser frequency and $f_{ref}$, as well as whether the laser frequency is higher or lower than $f_{ref}$.

The output of the MZI uses a 2 × 2 adiabatic broadband coupler[33] terminated to balanced photodetectors. As described in detail in Supplementary Note 1, the output error signal is asymmetric with respect to $f_{ref}$ and can be used as a servo signal to lock laser frequency. The error signal current is amplified and converted to voltage using a trans-impedance amplifier (TIA). The voltage signal is then fed into a proportional-integral-derivative (PID) controller, which modulates the laser current and corrects any frequency error. To compensate for

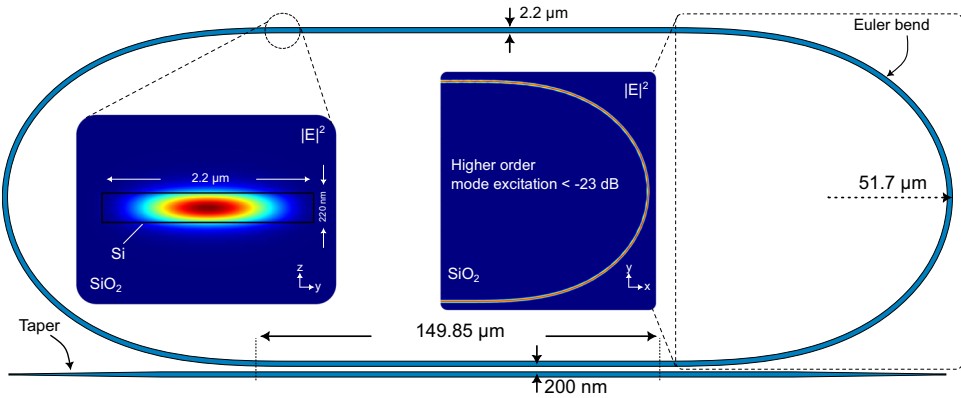

**Fig. 3 | High Q-factor silicon microring resonator.** In order to minimize the interface scattering loss due to waveguide edge roughness, a wide multi-mode waveguide is utilized. Moreover, to avoid the excitation of higher-order modes, an Euler bend is used. The finite-difference time-domain simulation of the waveguide cross-sections (y−z and x−y planes) shows the fundamental TE mode is well preserved inside the cavity.

potential fabrication-induced errors and adjust the microring resonance frequency and the optical phase of the bottom MZI branch, thermal phase shifters that are thermally isolated by a deep trench are utilized. Figure 2b shows the micro-photograph of the photonic integrated circuit implemented in the AIM Photonics 100 nm SOI process. The silicon photonic chip area is 0.456 mm$^2$.

## High Q-factor silicon microring resonator

Utilization of a microring resonator as an optical frequency reference plays a crucial role in OFND performance. A stable high Q-factor microring resonator, when coupled to the MZI, effectively enhances the sensitivity of the OFND. This, in turn, directly impacts the closed-loop operation and the ultimate laser frequency noise suppression. It is important to highlight that the proposed architecture can be implemented not only on various material platforms, such as low-loss silicon nitride but also using bench-top ultra-low expansion and stable etalons. Choosing silicon as a platform to implement the optical frequency reference is a trade-off between different design considerations such as potential co-integration with CMOS electronics, chip area, scalability, cost, and ultimate frequency stability.

In order to achieve a high Q-factor microring resonator, it is necessary to minimize intra-cavity losses. The propagation loss of silicon nanophotonic waveguides is influenced by several factors, with interface scattering and bend radiation loss being the most significant in a state-of-the-art silicon photonic foundry process[34–36]. The top-bottom surface roughness of a waveguide is well-controlled by the foundry, and it is not a design parameter. However, careful design of waveguide width can significantly improve the propagation loss. Figure 3 illustrates the implemented microring resonator in silicon. Utilizing multi-mode waveguides can reduce TE-mode overlap with side-wall roughness and greatly enhance the waveguide transmission[34]. The implemented multimode waveguide is 2.2 μm wide, and the theoretical fit model applied to the measured microring resonance response suggests a propagation loss of approximately 0.2 dB cm$^{-1}$. Although wide multi-mode waveguides can greatly reduce interfacial scattering loss, bend radiation loss increases significantly if a tight bend is used[37], especially for a highly multi-mode waveguide. An Euler bend is employed to achieve a compact multi-mode bend with minimal excitation of higher-order modes and mode cross-talk. The ring resonator is designed to achieve critical coupling to maximize OFND sensitivity. However, the fabricated ring is slightly under-coupled due to the potential fabrication-induced errors in the coupling region. Supplementary Note 3 discusses OFND gain sensitivity to waveguide loss and ring coupling ratio.

As illustrated in Fig. 3, the fundamental TE mode is preserved within the bent multi-mode waveguide with higher-order mode excitation of less than −23 dB. The implemented microring resonator has a circumference of about 950 μm, which corresponds to a free spectral range (FSR) of about 83.5 GHz with a loaded Q-factor of about $2.5 \times 10^6$ at the resonance wavelength of 1550.73 nm.

## The open loop operation: device characterization and the error signal

Figure 4a shows the schematic of the measurement setup to characterize the open loop performance and the error signal. A tunable external cavity diode laser (ECDL) with a wavelength of 1550.7 nm is coupled into the chip via the on-chip grating coupler. The narrow linewidth ECDL with linear frequency chirp facilitates accurate characterization and calibration of the ring resonator and the error signal responses. To ensure linear operation of the high Q-factor microring resonator, laser power is controlled using a variable optical attenuator (VOA). The silicon chip temperature is stabilized at 27 °C. The laser frequency is continuously scanned within the range of 30 GHz. A calibrated fiber-based MZI with an FSR of 20 MHz is used for time-frequency conversion. As shown in Fig. 4a, an on-chip sniffer photodetector with a 5% coupling ratio is utilized to monitor the resonance response. The resonance response of the ring and the error signal are measured simultaneously using an oscilloscope. Figure 4b shows the response of the ring where the Q-factor is about $2.5 \times 10^6$. The measured extinction ratio is about 4.8 dB which suggests the fabricated microring is slightly under-coupled, likely due to fabrication error in the coupling gap. Figure 4c shows the measured normalized asymmetric error signal that suggests OFND sensitivity of $1.38 \times 10^{-8}$ Hz$^{-1}$, which agrees well with analytical models. Heaters 1 and 2 are off during the measurement.

## The closed-loop operation and laser frequency noise suppression

Figure 5a shows the block diagram of the closed-loop measurement setup. As a proof-of-concept demonstration of laser frequency locking, given the thermo-refractive noise (TRN) of the integrated microring resonator, three different commercially available DFB lasers at wavelengths of 1550.7 nm, 1547.8 nm, and 1551.4 nm with large enough frequency noise are chosen. Moreover, the electronic noise of the laser bias current is adjusted and controlled to accentuate the laser frequency locking demonstration. As shown in Fig. 5a, 10% of the DFB laser power is passed through a VOA followed by a polarization controller and coupled into the chip via the on-chip grating coupler. This arrangement allows control over the coupled optical power within the range of 0.5 mW to 1 mW. The generated error signal is amplified using a low noise TIA with a gain of 5 kΩ and simulated total integrated input-referred current noise of 3.4 pA Hz$^{-1/2}$. Details of the TIA circuit and its

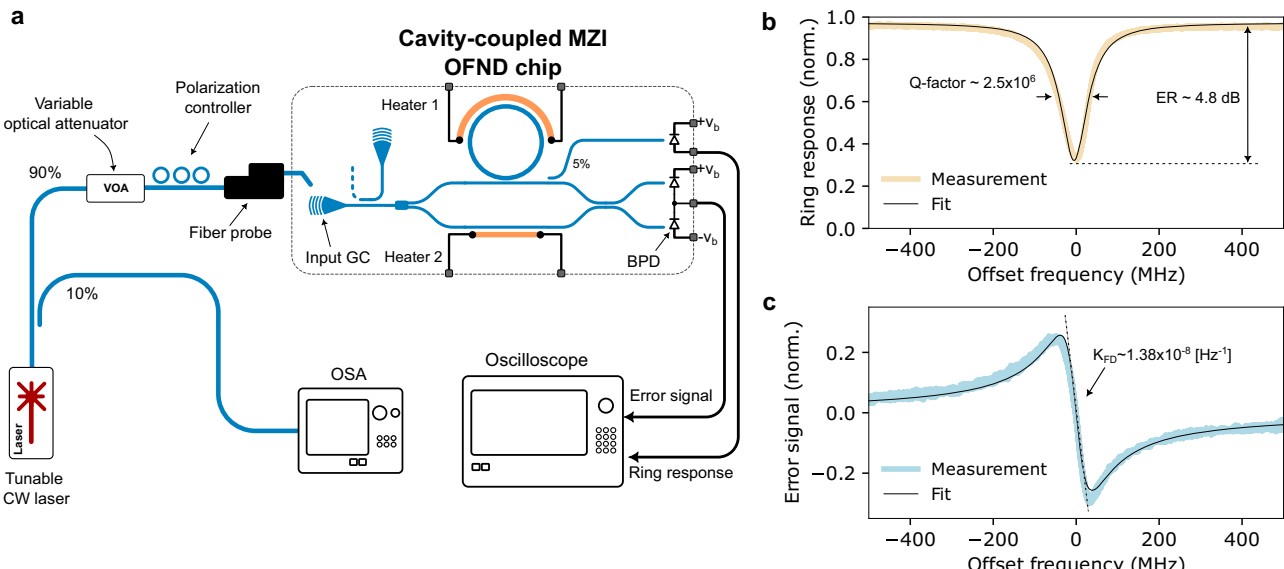

**Fig. 4 | The open loop response. a** Measurement setup for the open loop and microring resonator characterization. **b** The microring resonator response is measured via the on-chip photodetector sniffer. **c** The measured asymmetric error signal. Heaters are off during the test. ER extinction ratio, CW continuous wave, OSA optical spectrum analyzer, GC grating coupler, BPD balanced photodetector.

performance are presented in Supplementary Note 4. The output of the TIA is passed through a PID controller and also sampled with a field programmable gate array (FPGA) for in-loop frequency noise power spectral density (PSD) analysis. Analyzing in-loop frequency noise PSD retrieved from the in-loop error signal PSD provides valuable information that can be used to perform real-time adjustment of loop parameters, leading to improved frequency noise suppression performance and loop stability. Finally, the output of the PID controller modulates the laser diode current to adjust the laser frequency error.

The remaining laser power is used in heterodyne beat note measurement[18] using a fully stabilized octave-spanning optical frequency comb[38]. Details of the frequency noise measurement are explained in the Method section. Figure 5b shows the measured frequency noise PSD of the free-running and stabilized DFB laser one at the wavelength of 1550.7 nm. As shown in Fig. 5b, the cavity-coupled MZI OFND can effectively suppress the frequency noise of the free-running DFB laser by about 40 dB at 1.5 kHz Fourier frequency. We define the loop bandwidth as the frequency at which the frequency noise suppression factor becomes unity, and the loop no longer suppresses noise at higher frequencies. Figure 5b suggests that the root mean square (RMS) of laser frequency noise ($\sigma_{\delta f}$) is suppressed from 4.28 MHz to 400.3 kHz within the loop bandwidth of 80 kHz. Figure 5c shows the measured frequency noise PSD of DFB laser two, lasing at the wavelength of 1547.8 nm. As shown in Fig. 5c, the free-running laser frequency noise is suppressed by more than 42 dB at about 1.5 kHz Fourier frequency. Within the loop bandwidth of 100 kHz, $\sigma_{\delta f}$ of the free-running DFB laser two is reduced from approximately 3.87 MHz to about 240.3 kHz. Figure 5d shows the measured frequency noise PSD of the free-running and stabilized DFB laser three at the wavelength of 1551.4 nm. As shown in Fig. 5d, the frequency noise is suppressed by 39 dB at 1.5 kHz Fourier frequency and $\sigma_{\delta f}$ of laser frequency noise within the loop bandwidth of 150 kHz is suppressed from 1.7 MHz to 140.1 kHz.

The in-loop error signal PSD (Fig. 5a) reveals the relative frequency noise PSD between the laser and the microresonator, indicating negligible impact from absolute cavity TRN. Therefore, it can be used to estimate the frequency locking performance and ultimate frequency noise suppression, assuming the same loop parameter (e.g., gain and noise) but an ideal cavity with negligible TRN. As shown in Fig. 5b–d,

the in-loop frequency noise PSDs suggest that, indeed, the dominant limiting noise contribution in the loop is the microring TRN, and the proposed frequency stabilization loop is capable of suppressing the close-in frequency noise to less than $10^2\,\mathrm{Hz}^2\,\mathrm{Hz}^{-1}$ (extra 30 dB frequency noise suppression), emphasizing the effectiveness of the proposed approach.

When considering the integral linewidth, it's worth noting that in Fig. 5b–d, the $\beta$-separation line determines the part of the frequency noise PSD that contributes the most to the laser linewidth[39]. To effectively suppress a laser integral linewidth, the servo loop bandwidth must be greater than the frequency at which the $\beta$-line intersects with the frequency noise PSD. While the frequency-locked DFB1 and DFB2 demonstrate a significant reduction in close-in frequency noise, their integral linewidth is suppressed by only a factor of 2.1 and 2.9, respectively, attributed to a larger initial linewidth and a smaller servo bump bandwidth. In contrast, the integral linewidth of stabilized DFB3 drops from 4 MHz to 330 kHz (approximately a factor of 12), thanks to a higher servo bump frequency and a lower free-running Lorentzian linewidth (Fig. 5d). Supplementary Note 5 discusses the closed-loop operation in presence of noise sources and includes an example demonstrating the frequency noise PSD and linewidth estimation of a laser locked to a high-Q silicon nitride microresonator with significantly lower TRN.

## Discussion

Significant technological advancements have been implemented to develop state-of-the-art optical frequency references. These resonators can be categorized into two main groups: vacuum-gaped FP cavities[27,40–43] or dielectric microresonators[24,44]. Figure 6 demonstrates the comparison between different optical frequency reference technologies in terms of achievable TRN limit as a function of cavity volume. The minimum achievable TRN, as depicted in Fig. 6, is determined by the technology employed and the cavity volume. As shown by the trend line in Fig. 6, there is a clear trade-off between cavity size and the minimum achievable thermal noise. Ideally, a cavity that supports a larger mode volume and is made from a temperature-insensitive material can enable a lower TRN limit. As far as scalability and integration level are concerned, ultra-low loss silicon nitride[24,45] and micro-fabricated mirrors[41] present

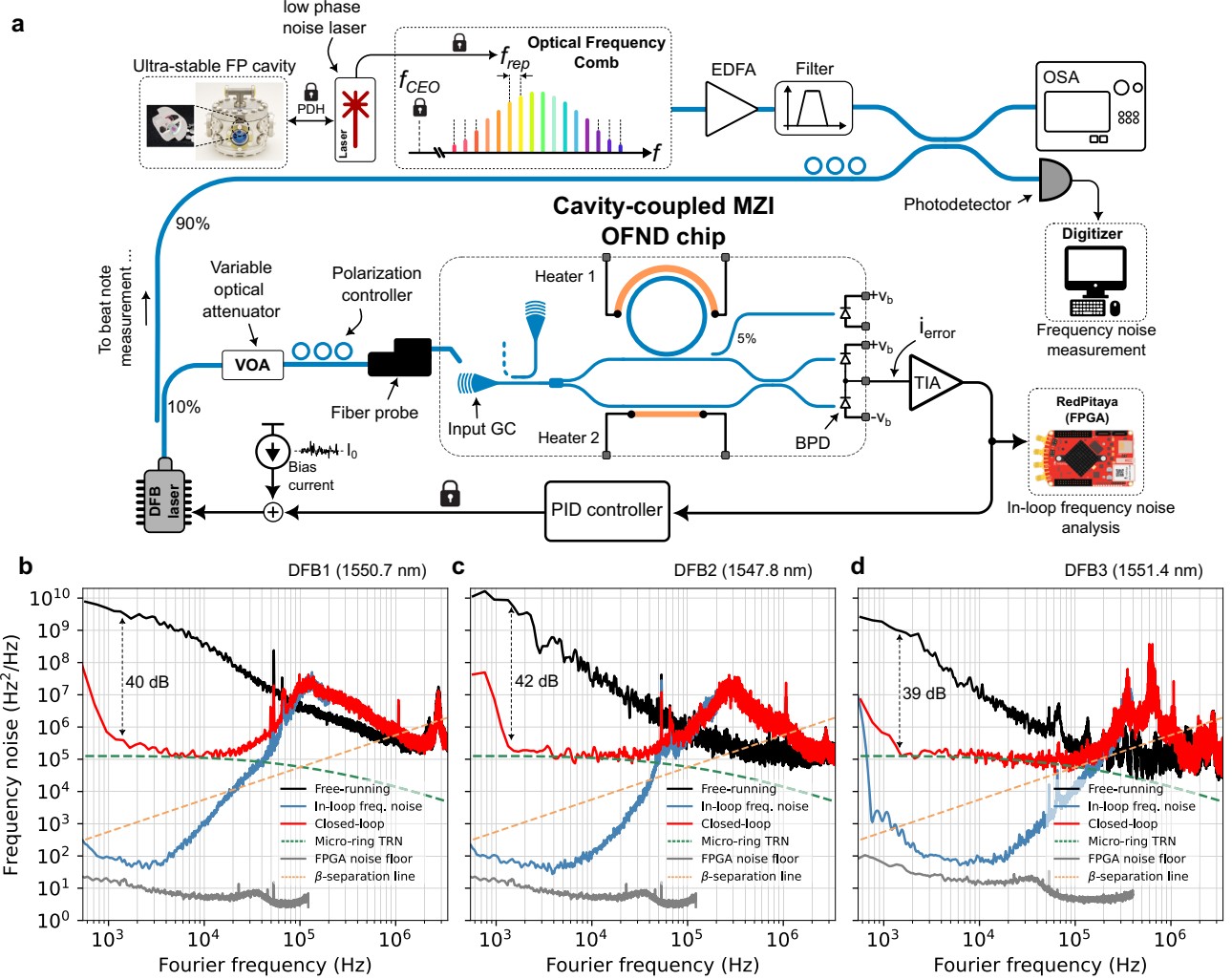

**Fig. 5 | The closed-loop operation. a** Measurement setup for the laser frequency locking experiment. Part of the laser output is beaten with the output of the fully-stabilized optical frequency comb and photodetected using a fast receiver. The beat note is then digitized and used for frequency noise analysis. Heaters are off during this measurement. **b**–**d** The power spectral density (PSD) of frequency noise of three different DFB lasers under free-running and closed-loop operation. The in-loop error signal is digitized and processed using an FPGA for in-loop frequency noise PSD estimation. The frequency noise of the stabilized DFB lasers is limited to the thermo-refractive noise (TRN) of the silicon microring. The RBW of laser frequency noise measurements is 200 Hz. RBW of in-loop noise measurements in (**b**) and (**c**) is 51 Hz, and in (**d**) is 103 Hz, respectively. **b**–**d** share the same vertical axis. EDFA Erbium-doped fiber amplifier, FP Fabry–Pérot, FPGA field programmable gate array.

themselves as promising technology platforms for fulfilling these requirements. However, this improvement comes at the cost of chip area, packaging complexity[46], and availability of the technology for large-scale and robust integrated photonic chip manufacturing. On the other hand, silicon photonic platforms are evolving rapidly, and while silicon may not be the most suitable choice for achieving highly stable micro-cavities, it certainly serves as an excellent alternative in numerous applications where a strict level of laser stability may not be required. In such cases, monolithic integration with CMOS devices and a compact footprint is often more desirable. The combination of robust large-scale silicon photonic manufacturing and the ability to monolithically integrate with mature CMOS electronics provide integrated silicon photonics a significant advantage over other alternatives. It is important to note that the choice of cavity technology and design is ultimately up to the designer and, indeed, application-specific. It is worth emphasizing that the proposed modulation-free laser stabilization method can be implemented on any suitable technology platform to effectively meet the requirements of an application. Supplementary Note 4 examines various noise sources, including the microresonator TRN.

Supplementary Note 5 explores the impact of these noise sources on closed-loop operation, and Supplementary Note 6 details system-level design procedures and considerations.

In conclusion, we have proposed and experimentally demonstrated a passive laser frequency noise stabilization method using integrated cavity-coupled MZI. The proposed method provides compact, scalable, and sensitive frequency noise discrimination without the need for electro-optic modulation, distinguishing it from the well-known PDH loop. The integrated cavity-coupled MZI consists of a high-Q Euler microring resonator with a loaded quality factor of about $2.5 \times 10^6$, an extinction ratio of about 4.8 dB, and a normalized OFND gain of $1.38 \times 10^{-8}\,Hz^{-1}$. As a proof-of-concept demonstration, the cavity-coupled MZI is used to suppress the frequency noise of three commercially available DFB lasers by about 40 dB at 1.5 kHz Fourier frequency. The implemented chip occupies 0.456 mm² area and is integrated on AIM Photonics, a commercially available silicon photonic process. The proposed architecture enables similar frequency noise measurement sensitivity compared to the well-known PDH method with simpler modulation-free architecture, which makes it easier to integrate into a single chip.

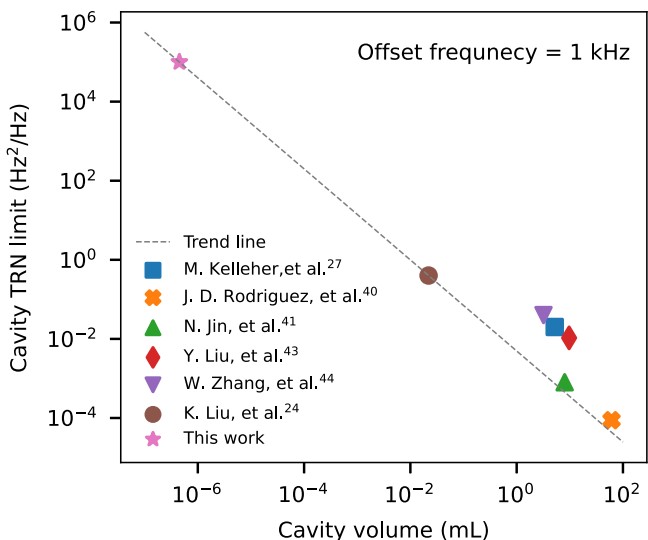

**Fig. 6 | Thermo-refractive noise in optical cavities.** The thermo-refractive noise of different cavity technologies at 1 kHz offset frequency shows a clear trade-off between the achievable noise floor and the cavity mode volume. As shown by the trend line, the TRN reduces as the cavity mode volume increases. Moreover, vacuum-gapped cavities such as refs. 27,40,41,43 show significantly lower TRN compared to their counterparts made from a dielectric at the cost of more complex packaging and implementation of a vacuum system. Detailed discussion and noise analysis are included in Supplementary Note 4.

## Methods

### Photonic chip implementation

All photonic devices are monolithically integrated on AIM Photonics 100 nm SOI process. The laser is coupled into the chip vertically using a grating coupler. A Y-junction with an excess loss of less than 0.5 dB is utilized at the input of the MZI to split the laser intensity equally. A high Q-factor microring resonator is coupled to the top arm of the MZI. The ring resonator is made with 2.2 μm wide multi-mode waveguides, and Euler bends to both reduce the interfacial scattering loss and avoid excitation of higher-order modes. Thermal phase shifters are integrated into the photonic chip for potential adjustment of phase or compensation for fabrication-induced errors. The two arms of the MZI are combined using an adiabatic broadband directional coupler. Balanced Germanium photodetectors are used to generate the error signal. The responsivity and dark current of the on-chip photodetector at 2.5 V reverse bias voltage are 1.16 A W$^{-1}$ (at 1550 nm) and 40 nA, respectively. The photonic chip occupies an area of 0.456 mm$^2$.

### The closed-loop operation in the presence of noise sources

Different noise sources that contribute to the ultimate frequency noise limit are discussed in Supplementary Note 4. Two main sources of noise are the TRN of the reference cavity and the total electronic noise, including input referred noise of electronics and the shot noise of balanced photodetectors. Supplementary Note 5 covers the suppression of frequency noise under closed-loop conditions and explores the minimum achievable frequency noise in the presence of various noise sources. As an example, Supplementary Note 5 includes a numerical simulation of the frequency noise PSD of a laser which is locked to a low TRN silicon nitride microresonator.

### Heterodyne beat note measurement

A low phase noise reference laser at the wavelength of 1564.68 nm with <7 Hz Lorentzian linewidth is PDH locked to the fundamental mode of an ultra-stable FP cavity made with ultra-low expansion spacers. A small portion of the stabilized laser output is used as the optical frequency reference to lock the octave-spanning fiber frequency comb (FFC) and stabilize the repetition rate to 200 MHz.

The FFC is self-referenced, and the carrier-envelope offset is also stabilized. The fully stabilized FFC is amplified using an optical amplifier and filtered with an optical band-pass filter tuned around the wavelength of DFB lasers under test. The DFB laser output is mixed with the FFC output, and the photodetected signal is filtered by a 98 MHz low-pass filter to reject the image beat note. The beat note is then digitized for frequency noise analysis. The list of all equipment and tools is provided in Supplementary Table 2.

### The OFND gain sensitivity analysis

The design of the high Q-factor microring resonator in the OFND directly impacts the frequency noise detection gain and laser frequency noise suppression. Supplementary Note 3 discusses the sensitivity of OFND gain to both the coupling ratio of the ring and the waveguide propagation loss.

## Data availability

All data supporting the findings of this study are available within the article and its supplementary files. Any additional requests for information can be directed to, and will be fulfilled by, the corresponding author.

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

## Acknowledgements

We thank Andrea Blanco-Redondo for helpful discussions and support.

## Author contributions

M.H.I. conceived the project idea. M.H.I. designed, simulated and laid out the integrated photonic circuits. M.H.I designed the printed circuit board and the on-board electronics. M.H.I. conducted measurements, data analysis, and simulations. K.K. packaged the integrated photonic chip. N.K.F. helped with frequency noise measurement. M.H.I. wrote the paper.

## Competing interests

The authors declare no competing interests.
