## [Peer Review File · Nature Communications]

REVIEWER COMMENTS

Reviewer #1 (Remarks to the Author):

The proposed method/architecture for laser stabilization in the submitted paper is of significant importance for various applications such as optical atomic clocks and microwave generation. These applications strongly depend on the laser linewidth, where fully integrated PDH locking, especially without the need for modulation, is very beneficial. I can strongly recommend the manuscript for publications in Nature Communications. Below are my comments on the submitted manuscript.

1. Very recently, a few works on microwave generation in a fully integrated system have been demonstrated. Those works would benefit from the proposed architecture and, in my opinion, they deserve to be mentioned in the introduction of the submitted manuscript to signify the importance of the work.

Kudelin, Igor, et al. "Photonic chip-based low noise microwave oscillator." arXiv preprint arXiv:2307.08937 (2023).

Sun, Shuman, et al. "Integrated optical frequency division for stable microwave and mmWave generation." arXiv preprint arXiv:2305.13575 (2023).

2. How the noise of the MZI itself affect the stability of the locking? Since the MZI is 'in-loop', the noises of the MZI would be transferred to the CW laser. It would be nice to see the SNR of the system. And it would be informative to know why the frequency noise do not reach the TRN.

3. The comparison to self-injection locked (SIL) lasers should be provided, since SIL approach is a simple realization of linewidth reduction.

4. One of my biggest concerns is the delay line for the self-heterodyne measurement: 27 meter can be not enough. In Fig.5(b) there is no servo bump and the frequency noise at higher offset frequencies does not merge with the free-running performance as it should be.

5. At page 2 Authors said 'Despite excellent performance, the PDH scheme requires electro-optic phase modulator and relatively fast and complex electronics for modulation and demodulation which increases the power consumption and area for an integrated PDH chip [19, 20].' I disagree that PDH requires high speed electronics, since it can operate at 10 MHz and below. Similar comment to the page 'Figure 1(b) shows the proposed cavity-coupled MZI as an OFND that maintains a simple passive structure without any need for fast optical phase modulation.' – 10 MHz is not considered as high electronics.

6. On page 7: 'The ring resonator is designed to achieve critical coupling to maximize OFND sensitivity, however, the fabricated ring are slightly under-coupled due to the potential fabrication-induced errors.' – 'are' should be changed to 'is'.

Reviewer #2 (Remarks to the Author):

- What are the noteworthy results?

The authors propose a novel, modulation-free laser stabilization method. To the best of my knowledge the method is indeed novel, and this is perhaps the strongest aspect of this manuscript. The authors also demonstrated the same method on a SOI platform with reasonably good results. The authors state that the obtained result is close to the thermorefractive noise limit (though I would strongly suggest adding the TRN calculation in the supplement).

I believe the novelty of the method and the fact that it is accompanied by a demonstration makes this manuscript appropriate for publication in Nature Communications. However, there are several important revisions that are needed to better place this method and demonstration in the proper context, these are outlined below approximately answering the questions posed by the editor.

- Will the work be of significance to the field and related fields? How does it compare to the established literature? If the work is not original, please provide relevant references.

Mostly yes.

On the one hand the demonstrated frequency discriminator provides a very compact and low complexity approach to stabilizing a laser to a micro-ring resonator. On the other hand, the results obtained can be surpassed with even a free-running fiber laser. However, it is true that the footprint of the demonstrated chip is small and that fiber lasers are not available at all relevant wavelengths (whereas diodes can be made at a myriad of wavelengths). In any case, I think it is important that the authors acknowledge and compare their device and method to other existing technologies and thus they can provide a more complete picture to the reader. Two examples: (1) as mentioned above, free-running fiber lasers have linewidths in the few kHz level, more than two orders of magnitude better than the demonstrated ~ 700 kHz. (2) External cavity diode lasers (ECDL) provide linewidths at the 100 kHz level. The authors could fairly emphasize the main advantages of their method which I believe are area/volume and perhaps power consumption - though this is a nuanced situation given that the control loop in this demonstration is a lab PID unit and may have a significant power draw.

The authors compare their method to PDH but relevant trade-offs are overlooked in the manuscript. Namely, (1) the PDH technique is well-known and widely adopted because it has low error signal offsets and thus it can provide a high-quality lock to the peak of the desired resonance (well known sources of shifts are residual reflections and/or residual amplitude modulation at the PDH frequency). In the method proposed here, the slope of the error signal seems to depend on the phase (ϕ in their equations) of the two fields interfering at the output of the Mach-Zehnder interferometer. (2) Similarly, PDH is insensitive to laser amplitude noise because the phase/frequency error information is moved to a high frequency band (the PDH modulation frequency) and then coherently demodulated. A homodyne interferometer downconverts all of the amplitude noise of the laser and this should be mentioned as well.

Another point: of the applications mentioned in the introduction, it is difficult to believe that this specific demonstration - at least in its current form - will have an impact on optical atomic clocks as the laser linewidths required for a clock interrogation laser are at between 4 and 6 orders of magnitude better than demonstrated here.

At the end of the manuscript the authors provide routes for reducing the TRN such as using a SiN ring or ultralow loss Si. This is important; however, this comparison requires some numbers. What kind of

improvement in frequency noise or linewidth is expected? At what volume/area cost? Similarly, for a macroscopic cavity: the authors could give one example of the noise improvement and corresponding system volume increase if this method were implemented on a standard stable reference cavity. I don't believe that simply saying that they would provide better performance at the cost of higher chip area provides the reader enough information.

- Does the work support the conclusions and claims, or is additional evidence needed?

Mostly. A few comments though:

One interesting question is the "line-splitting" factor achieved in this demonstration. The resonance width is 80 MHz and final linewidth 700 kHz, implying a line-splitting of ~100x. However, if the limitation is the TRN of the ring, as the authors state, this is a limitation of the device, not a limitation of the method. It would be interesting if the authors showed an in-loop error signal PSD measurement to show that the method would be able to support references with lower thermal noise. Essentially, I am asking for a measurement of the residual noise of the lock $(S_{0}(f)/(K_{E} * K_{L} * K_{FD})^2 + S_n(f)/K_{FD}^2)$, as this could show the power of this method if it were applied to a different, better resonator and would strengthen the manuscript.

The authors did the initial characterization of their ring with an ECDL and then they switched to a DFB laser for the lock demonstration. The reason for this change is not stated though I suspect this is because the linewidth of the ECDL was too narrow to show an improvement in the demonstration experiment. If this is the case, I understand that the demonstration requires a laser that is bad enough such that it can be *improved* with the method, but this is a point that needs to be explicitly acknowledged so the reader does not have to guess.

- Are there any flaws in the data analysis, interpretation and conclusions? - Do these prohibit publication or require revision?

Not a major flaw, but this manuscript would benefit a few-sentence and few-equation supplement showing the TRN calculation. This way the reader can double-check and/or use the same methodology to calculate the TRN for a different ring.

- Is the methodology sound? Does the work meet the expected standards in your field?

Generally, yes.

- Is there enough detail provided in the methods for the work to be reproduced?

The method itself can likely be reproduced in another experiment with the information available. I do not have enough expertise on device fabrication to say whether there is enough information to make the same device.

-Some typos:

In the first introduction paragraph:

"The Optical feedback method relies "  Perhaps there is no need to capitalize optical

In the last sentence of section I:

“The proposed architecture offers a promising solution for achieving sensitive, simple, and low power”
 simple

In section II(D):

“A tunable continuous wave laser (TOPTICA CTL 1550)”  continuous

-Other minor comments:

“ $|T(\omega)|$, $\psi(\omega)$, ϕ , P_0 , and R are the amplitude and phase of the optical reference at the frequency of ω ” -
-> Amplitude and phase of the reflection coefficient of the optical reference.

Respond to reviewers comments

First and foremost, we would like to thank all the reviewers for their time and valuable comments that have certainly improved our manuscript. Please find our point by point response to all comments below. Our responses are shown in **blue** and the changes made in the revised manuscript are shown in **red**. Please note that we have added a list of all major changes at the end of this document.

Reviewer #1 (Remarks to the Author):

The proposed method/architecture for laser stabilization in the submitted paper is of significant importance for various applications such as optical atomic clocks and microwave generation. These applications strongly depend on the laser linewidth, where fully integrated PDH locking, especially without the need for modulation, is very beneficial. I can strongly recommend the manuscript for publications in Nature Communications. Below are my comments on the submitted manuscript.

We sincerely appreciate the valuable comments provided by the reviewer, which have strengthened our manuscript. We believe that our proposed modulation-free laser stabilization method can have a significant impact in the community and in applications where laser stability is critical. We tried to prepare the majorly revised manuscript along with the detailed Supplementary Information document to facilitate the repeatability of our work and assist readers in implementing our proposed architecture in various systems.

Below, we address the comments and questions point-by-point.

1. Very recently, a few works on microwave generation in a fully integrated system have been demonstrated. Those works would benefit from the proposed architecture and, in my opinion, they deserve to be mentioned in the introduction of the submitted manuscript to signify the importance of the work.

Kudelin, Igor, et al. "Photonic chip-based low noise microwave oscillator." arXiv preprint arXiv:2307.08937 (2023).

Sun, Shuman, et al. "Integrated optical frequency division for stable microwave and mmWave generation." arXiv preprint arXiv:2305.13575 (2023).

Thank you for your suggestions. We have added these references (ref. 8, 9) to the main manuscript. In light of reviewer comment, we have added the following paragraph to the main manuscript (page 1, line 1):

"Precise laser frequency control is a crucial requirement in various applications, including optical communication [1, 2], optical atomic clocks [3, 4], sensing [5, 6], and photonic-assisted ultra-low phase noise microwave and mmWave synthesis [7–9], which makes stable and narrow linewidth lasers indispensable part of precision optical experiments."

2. How the noise of the MZI itself affect the stability of the locking? Since the MZI is 'in-loop', the noises of the MZI would be transferred to the CW laser. It would be nice to see the SNR of the system.

Thank you for your comments and questions.

For the sake of analysis, we decoupled "microresonator TRN" from "MZI noise". In other words, when analyzing "MZI noise" we consider an ideal cavity with zero TRN, and when considering cavity TRN, we assume MZI to be ideal.

With that said, as the reviewer suggested, MZI noise should be generally considered, however, in our integrated chip, since the device sizes are orders of magnitude smaller than, say fiber based MZI, the MZI noise is negligible compared to other sources of noise (e.g. microresonator TRN, photodetector shot noise, and input referred noise of electronics). As a numerical example in Supplementary Note 4, the rms current noise due to phase noise in MZI (i.e. MZI noise) made by a 500 μm long single-mode silicon waveguide is about $7 \text{ fA}\cdot\text{Hz}^{-1/2}$. On the other hand the total rms electronic noise including photodetector shot noise and TIA input referred current noise is about $5.2 \text{ pA}\cdot\text{Hz}^{-1/2}$ which is about 3 orders of magnitude larger than the MZI noise. This is thanks to the miniaturization of the nanophotonic waveguides and MZI. We would like to make this point clear that the microresonator TRN is amplified by the OFND gain (see Supplementary Equation (33) in Supplementary Note 5) which is because of the MZI arrangement and technically can be considered noise due to MZI but we call it cavity TRN throughout the manuscript.

In light of reviewer's comment, we have prepared Supplementary Note 4 where different noise sources, including MZI noise, and their contribution in closed-loop performance have been analytically and numerically discussed (page 9- Supplementary Information):

"As mentioned earlier, a waveguide with length of L at temperature T_0 will induce phase noise to the propagating optical wave due to thermo-refractive fluctuations. If such waveguide is used in an MZI arrangement followed by photodetector, the phase noise will translate into electronic amplitude noise. ..."

And it would be informative to know why the frequency noise do not reach the TRN.

Thank you for your comment. In light of the reviewers comments, we have repeated the closed-loop experiment for three different DFB lasers and measured frequency noise with a new method using fully stabilized octave-spanning fiber frequency comb and heterodyne beat note technique. As shown in Fig. 5 of the main manuscript, the frequency noise of the stabilized laser is indeed limited to the microresonator TRN. Please see the copied Fig. 5 on page 4 on the current document.

3. The comparison to self-injection locked (SIL) lasers should be provided, since SIL approach is a simple realization of linewidth reduction.

Thank you for your suggestion. We have added a paragraph on the comparison to self-injection locking scheme (page 1, starting on line 6).

"...In optical feedback method, or self-injection locking (SIL), a small portion of the laser output is filtered and injected back into its cavity [16, 17]. SIL method is a promising way to suppress laser

frequency noise across a large bandwidth. Although this method offers excellent short-term stability, frequency noise reduction over a large bandwidth, and significant linewidth suppression, it typically cannot guarantee long-term stability and requires active stabilization [18–20]. Moreover, designing the optical feedback usually mandates the co-design of laser system and optical signals with an ultra-high quality factor (Q-factor) cavity. Electro-optic techniques, on the other hand, leverage state-of-the-art and mature electronic devices and systems to precisely control the laser frequency and offer long-term laser frequency stabilization. This will push some of the challenges from optical domain into the electrical one where control and manipulation of the signals and systems are comparatively more manageable and cost-effective.”

4. One of my biggest concerns is the delay line for the self-heterodyne measurement: 27 meter can be not enough.

Thank you for your comment. We understand reviewer’s concern about measurement sensitivity and SNR, hence, in light of reviewer’s comment we have changed our frequency noise measurement setup to heterodyne beat note measurement (please see the image below). In our new setup, a narrow linewidth reference laser (< 7 Hz Lorentzian linewidth) is PDH locked to an ultra-stable FP cavity made from ULE spacer held rigidly inside a vacuum chamber to make a low noise optical frequency reference at wavelength of 1564.68 nm. To transfer the frequency stability across the C-band, an octave-spanning self-referenced fiber frequency comb is locked to the optical frequency reference. The fully stabilized comb with 200 MHz repetition rate is then used as reference in heterodyne beat note measurements. The beat note between DFB laser under test and the comb line is photodetected, digitized and used for frequency noise analysis (please see Fig. 5 in main manuscript). We have repeated laser frequency locking experiment for three different DFB lasers and have reported the results in Fig. 5(b-d).

In light of reviewer’s comments, we have changed the closed-loop operation and laser frequency noise suppression section accordingly (pages 9-12).

In Fig.5(b) there is no servo bump and the frequency noise at higher offset frequencies does not merge with the free-running performance as it should be.

Thank you for your comments and valuable feedback. As we explained earlier, we have changed our frequency noise measurement technique and used fully-stabilized frequency comb to measure laser frequency. Moreover, we have repeated the close-loop experiment using three different commercially available DFB lasers at 1550.7 nm, 1547.8 nm, and 1551.4 nm to fully study and demonstrate the performance of our proposed laser frequency locking method. As can be seen in Fig. 5(b-d), the servo bump is clear around 100 kHz, 300 kHz, and 600 kHz respectively for different DFBs and their corresponding PID parameters.

In light of reviewer’s comments, we have changed the closed-loop operation and laser frequency noise suppression section accordingly (page 9).

Fig. 5. The closed-loop operation. **a** Measurement setup for laser frequency locking experiment. Part of the laser output is beat with the output of the fully-stabilized optical frequency comb and photodetected using a fast receiver. The beat note is then digitized and used for frequency noise analysis. Heaters are off during this measurement. **b-d** The power spectral density of frequency noise of three different DFB lasers under free-running and closed-loop operation. The in-loop error signal is digitized and processed using an FPGA for in-loop frequency noise power spectral density (PSD) estimation. The frequency noise of the stabilized DFB lasers are limited to thermorefractive noise (TRN) of the silicon microring. The RBW of laser frequency noise measurements is 200 Hz. RBW of in-loop noise measurements in **(b)** and **(c)** is 51 Hz, and in **(d)** is 103 Hz, respectively. **(b-d)** share the same vertical axis. EDFA: Erbium-doped fiber amplifier, FP: Fabry-Perot, FPGA: field programmable gate array.

5. At page 2 Authors said ‘Despite excellent performance, the PDH scheme requires electro-optic phase modulator and relatively fast and complex electronics for modulation and demodulation which increases the power consumption and area for an integrated PDH chip [19, 20].’ I disagree that PDH requires high speed electronics, since it can operate at 10 MHz and below. Similar comment to the page ‘Figure 1(b) shows the proposed cavity-coupled MZI as an OFND that maintains a simple passive structure without any need for fast optical phase modulation.’ – 10 MHz is not considered as high electronics.

Thank you for your comments. We agree with the reviewer that 10 MHz is not considered as high in electronics. However, the PDH modulation frequency should be chosen given the cavity

resonance linewidth for optimal operation. The higher the cavity finesse, the lower the optimum modulation frequency. For the sake of argument, in our comparison in Fig. 1(c), we are assuming a cavity made in a conventional integrated silicon photonics process (page 5, line 10):

“To assume a fixed area to implement the OFND, a 1 mm circumference ring resonator is used as the frequency reference in both the PDH and the cavity-coupled MZI. The length mismatch between the arms of the unbalanced MZI is also set to 1 mm. The waveguide loss is assumed 0.2 dB.cm⁻¹.” With the given numbers, the modulation frequency chosen in simulation of Fig. 1(c) is 1 GHz. Please see Supplementary Note 2 for more details.

We are trying to convey the point that regardless of the quality of the reference cavity, our proposed locking architecture offers similar frequency discrimination sensitivity as the PDH scheme if they both use the same reference. The PDH requires modulation and de-modulation at a MHz-GHz frequency whereas our proposed structure does not. This will significantly affect complexity and power consumption of system. Moreover, implementing efficient phase modulation in integrated platforms that offer high-Q cavities, like silicon nitride, is not trivial which makes our proposed architecture well-suited for those platforms too.

Despite the pros of our proposed architecture, it comes at a cost. In PDH loop, due to this modulation/demodulation it is normally less vulnerable to DC error offset and unwanted noise at base-band. To mitigate this in our architecture a careful design needs to be considered.

In light of the reviewer’s comments, we have added these sentences to the main manuscript:

- Page 2, line 20:

“Despite excellent performance, the PDH scheme requires electro-optic phase modulator, and fast enough electronics for modulation and demodulation which increases the power consumption and area for an integrated PDH chip [29, 30]. It is worth mentioning that the properties of the electronics in the PDH loop such as modulation speed, amplitude, and trans-impedance gain depend on the optical frequency reference characteristics such as the cavity finesse. As a result, depending on the cavity finesse, the modulation speed may vary from MHz to GHz range. Moreover, as far as integration level is concerned, ultra-high Q-factor integrated cavities are usually implemented in platforms (e.g. silicon nitride) that lack efficient electro-optic phase modulation in C-band. This makes full integration of PDH system in a single photonic chip challenging.”

- Page 5, paragraph 2:

“To provide a fair comparison, although the proposed cavity-coupled MZI architecture offers a simple and sensitive frequency noise measurement technique, it demands careful design consideration. The PDH technique is less vulnerable to DC offset error and base-band noise, however, the frequency content of the proposed OFND is at the base-band and more susceptible unwanted low frequency fluctuations which demands careful noise and link budget analysis (see Supplementary Notes 4). Also, the slope of the OFND response depends on ϕ in Eq. (1) and has to be set properly. The random phase variation can be significantly mitigated by miniaturizing the system on a chip within a compact footprint. This results in a small temperature gradient across the integrated chip, thereby suppressing random phase

difference fluctuations caused by temperature changes. For instance, in the closed-loop laser frequency locking experiments we observed a steady and robust lock without active MZI phase locking with on-chip thermal phase shifters turned off. Supplementary Note 2 describes the comparison between different OFND architectures.”

6. On page 7: ‘The ring resonator is designed to achieve critical coupling to maximize OFND sensitivity, however, the fabricated ring are slightly under-coupled due to the potential fabrication-induced errors.’ – ‘are’ should be changed to ‘is’.

Thank you for your comment. We have corrected this typo.

Reviewer #2 (Remarks to the Author):

We would like to thank the reviewer for their valuable comments and feedback which have certainly made our manuscript stronger. Below, we address the comments and questions point-by-point.

- What are the noteworthy results?

The authors propose a novel, modulation-free laser stabilization method. To the best of my knowledge the method is indeed novel, and this is perhaps the strongest aspect of this manuscript. The authors also demonstrated the same method on a SOI platform with reasonably good results. The authors state that the obtained result is close to the thermorefractive noise limit (though I would strongly suggest adding the TRN calculation in the supplement).

Thank you for your comments and suggestions. We believe our novel laser frequency stabilization method can have a significant impact in both the field and applications where a simple, yet effective technique is crucial. To assist readers in replicating our results and designing their own systems based on our proposed technique, we have prepared an extensive Supplementary Information document to go along with our majorly revised manuscript.

In light of reviewer's suggestion, we have added Supplementary Note 4 in which different noise sources are discussed, including thermorefractive noise (pages 6-11, Supplementary Information).

I believe the novelty of the method and the fact that it is accompanied by a demonstration makes this manuscript appropriate for publication in Nature Communications. However, there are several important revisions that are needed to better place this method and demonstration in the proper context, these are outlined below approximately answering the questions posed by the editor.

We would like to thank the reviewer for valuable comments, suggestions, and feedback which has certainly improved our work.

- Will the work be of significance to the field and related fields? How does it compare to the established literature? If the work is not original, please provide relevant references.

Mostly yes.

On the one hand the demonstrated frequency discriminator provides a very compact and low complexity approach to stabilizing a laser to a micro-ring resonator. On the other hand, the results obtained can be surpassed with even a free-running fiber laser. However, it is true that the footprint of the demonstrated chip is small and that fiber lasers are not available at all relevant wavelengths (whereas diodes can be made at a myriad of wavelengths).

Thank you for your comment. We fully agree that there are currently commercially available laser technologies that, although free running, exhibit very narrow linewidth (<10 kHz or even lower). However, we would like to strongly emphasize that we are not proposing a new laser technology;

we are proposing a new method that can improve both integrated and bench-top state-of-the-art laser locking systems. Similar to the well-known PDH or other laser frequency locking methods, our proposed approach is applicable to any laser in an appropriate setting. When paired with a carefully chosen optical frequency reference, it effectively suppresses laser frequency noise. We are demonstrating this as a proof-of-concept in a commercially available SOI chip to show the promise. We are aware of limitations and promise of this material platform, like any other integrated photonic technologies, and as discussed in the Discussion section, the choice of material platform and cavity is a design problem and, indeed, application specific.

In other words, one may decide to pick a cheap laser off-the-shelf with MHz-linewidth and use our proposed architecture in a silicon chip and leverage cost, scalability, area, and volume metrics but in another application the same proposed cavity-coupled MZI architecture can be implemented in an ultra-low loss silicon nitride technology, or even using free-space optics to lock free-running narrow linewidth lasers with kHz linewidth to achieve Hz-level final spectral linewidth. Here, we are using three different commercially available lasers to show a proof-of-concept demonstration of our proposed laser frequency noise suppression method.

In light of the reviewer comment, we have added this following sentences:

- Page 3, line 21:
“...We emphasize that our proposed laser stabilization technique is versatile and, similar to other laser frequency locking techniques (e.g., PDH), it can be applied across various platforms with a properly chosen frequency reference to achieve laser frequency noise suppression.”
- Page 13, line 14:
“...It is important to note that the choice of cavity technology and design is ultimately up to the designer and, indeed, application specific. It is worth emphasizing that the proposed modulation-free laser stabilization method can be implemented on any suitable technology platform to effectively meet the requirements of an application. Supplementary Note 4 examines various noise sources, including the microresonator TRN. Supplementary Note 5 explores the impact of these noise sources on closed-loop operation, while Supplementary Note 6 details system level design procedures and considerations.”

In any case, I think it is important that the authors acknowledge and compare their device and method to other existing technologies and thus they can provide a more complete picture to the reader. Two examples: (1) as mentioned above, free-running fiber lasers have linewidths in the few kHz level, more than two orders of magnitude better than the demonstrated ~700 kHz. (2) External cavity diode lasers (ECDL) provide linewidths at the 100 kHz level.

Thank you for your comments. As explained in the previous point, since we are not proposing a new laser we cannot make a fair comparison between our work and other existing laser technologies. However, we have made a clear comparison between our method and other frequency stabilization methods such as injection locking and the PDH loop. Moreover, as requested by reviewer's later questions, we have made a clear comparison between different optical frequency reference technologies in terms of TRN and cavity volume to make the trade-off clear enough. The choice of the optical frequency reference is crucial in determining the

ultimate frequency stability achieved, however, regardless of the designer's choice, our proposed architecture can be applied to attain laser stability up to the limits imposed by the chosen optical frequency reference.

To make this clear, the following sentences have been added to the main manuscript:

- Page 1, line 6:

“In optical feedback method, or self-injection locking (SIL), a small portion of the laser output is filtered and injected back into its cavity [16, 17]. SIL method is a promising way to suppress laser frequency noise across a large bandwidth. Although this method offers excellent short-term stability, frequency noise reduction over a large bandwidth, and significant linewidth suppression, it typically cannot guarantee long-term stability and requires active stabilization [18–20]”

- Page 2, line 14:

“Using the PDH technique, a sharp asymmetric error signal can be generated, which can then be utilized as a servo signal to stabilize the laser frequency. In PDH architecture, the laser phase is modulated using a phase modulator, filtered by an optical frequency reference (e.g. Fabry-Perot (FP) cavity) followed by a photodetector [28]. The photodetected signal is down converted with the same local oscillator frequency to base-band. This up/down conversion scheme makes PDH more robust against DC error signal offset, base-band noise, and other perturbations at low frequency which enables a robust stable frequency lock. Despite excellent performance, the PDH scheme requires electro-optic phase modulator, and fast enough electronics for modulation and demodulation which increases the power consumption and area for an integrated PDH chip [29, 30]. It is worth mentioning that the properties of the electronics in the PDH loop such as modulation speed, amplitude, and trans-impedance gain depend on the optical frequency reference characteristics such as the cavity finesse. As a result, depending on the cavity finesse, the modulation speed may vary from MHz to GHz range. Moreover, as far as integration level is concerned, ultra-high Q-factor integrated cavities are usually implemented in platforms (e.g. silicon nitride) that lack efficient electro-optic phase modulation in C-band. This makes full integration of PDH system in a single photonic chip challenging.”

- Page 5, paragraph 2:

“To provide a fair comparison, although the proposed cavity-coupled MZI architecture offers a simple and sensitive frequency noise measurement technique, it demands careful design consideration. The PDH technique is less vulnerable to DC offset error and base-band noise, however, the frequency content of the proposed OFND is at the base-band and more susceptible unwanted low frequency fluctuations which demands careful noise and link-budget analysis (see Supplementary Notes 4). Also, the slope of the OFND response depends on ϕ in Eq. (1) and has to be set properly. The random phase variation can be significantly mitigated by miniaturizing the system on a chip within a compact footprint. This results in a small temperature gradient across the integrated chip, thereby suppressing random phase difference fluctuations caused by temperature changes. For instance, in the closed-loop laser frequency locking experiments we observed a steady and robust lock without active MZI

phase locking with on-chip thermal phase shifters turned off. Supplementary Note 2 describes the comparison between different OFND architectures.”

- Page 12, 13 (Discussion section) including Fig. 6.

Fig. 6. Thermorefractive noise in optical cavities. Cavity TRN limit for different optical cavities at 1 kHz offset frequency shows a clear trend where TRN reduces as the cavity mode volume increases.

The authors could fairly emphasize the main advantages of their method which I believe are area/volume and perhaps power consumption - though this is a nuanced situation given that the control loop in this demonstration is a lab PID unit and may have a significant power draw.

Thank you for your comments. We agree with the reviewer that using a lab bench-top PID may have a significant power consumption. We acknowledge that to optimize the power consumption, the electronics need to be co-designed and integrated with the photonics for a given application. We have already demonstrated such electronic-photonic co-design in our previous works:

Idjadi, Mohamad Hossein, and Firooz Aflatouni. "Nanophotonic phase noise filter in silicon." Nature Photonics 14.4 (2020): 234-239.

Idjadi, Mohamad Hossein, and Firooz Aflatouni. "Integrated Pound–Drever–Hall laser stabilization system in silicon." Nature communications 8.1 (2017): 1209.

However, the focus of our work here is not demonstrating a fully integrated electronic-photonic demonstration of laser locking system though it certainly is an interesting topic to pursue in the future. Here, using a lab PID is just for proof-of-concept demonstration of the system as implementing an electronic circuit with simple low frequency amplifiers and filters is not challenging. In light of reviewer’s comment, we added the following sentence to the main manuscript (page 3, paragraph 3):

“...The proposed method can be co-integrated with mature electronics [14, 30], which sets the stage for the development of low-cost, scalable, and stable fully integrated lasers.”

The authors compare their method to PDH but relevant trade-offs are overlooked in the manuscript. Namely, (1) the PDH technique is well-known and widely adopted because it has low error signal offsets and thus it can provide a high-quality lock to the peak of the desired resonance (well known sources of shifts are residual reflections and/or residual amplitude modulation at the PDH frequency). In the method proposed here, the slope of the error signal seems to depend on the phase (ϕ in their equations) of the two fields interfering at the output of the Mach-Zehnder interferometer.

Thank you for your comment. As correctly noted by the reviewer, the phase of the interferometer (ϕ) affects the error signal. With the chip temperature stabilized and the entire system miniaturized, we did not actively phase-lock the MZI. In fact, both heaters 1 and 2 were off during the experiment. The laser frequency lock remained stable throughout several hours of measurement. However, we agree that if a designer chooses to implement our work using, for example, optical fibers then slow DC error drift might be an issue and MZI phase locking may be considered.

Integrating the proposed architecture into small footprints will significantly reduce random phase drift (due to thermal fluctuations) since temperature gradient across the chip is very low. In any case, having a slow thermal phase tuner, like the integrated thermal phase shifters, can be useful in situations where static or dynamic phase adjustment is needed.

In light of reviewer's comment, we have added the following paragraph to address these trade-offs in page 5, paragraph 2:

“To provide a fair comparison, although the proposed cavity-coupled MZI architecture offers a simple and sensitive frequency noise measurement technique, it demands careful design consideration. The PDH technique is less vulnerable to DC offset error and base-band noise, however, the frequency content of the proposed OFND is at the base-band and more susceptible unwanted low frequency fluctuations which demands careful noise and link-budget analysis (see Supplementary Notes 4). Also, the slope of the OFND response depends on ϕ in Eq. (1) and has to be set properly. The random phase variation can be significantly mitigated by miniaturizing the system on a chip within a compact footprint. This results in a small temperature gradient across the integrated chip, thereby suppressing random phase difference fluctuations caused by temperature changes. For instance, in the closed-loop laser frequency locking experiments we observed a steady and robust lock without active MZI phase locking with on-chip thermal phase shifters turned off. Supplementary Note 2 describes the comparison between different OFND architectures.”

(2) Similarly, PDH is insensitive to laser amplitude noise because the phase/frequency error information is moved to a high frequency band (the PDH modulation frequency) and then coherently demodulated. A homodyne interferometer downconverts all of the amplitude noise of the laser and this should be mentioned as well.

Thank you for your comments and valuable feedback. We indeed agree that one of the main advantages of the PDH scheme is that the laser phase noise content is upconverted to the frequency of modulation which will bypass $1/f$ flicker noise of electronics and other unwanted noise at base-band. We also think the choice of the architecture is ultimately a design problem

for a specific application. For example, the TRN of an available cavity (in a given chip area for a given material platform) may or may not be larger than the flicker noise at a frequency offset of interest so the designer may choose one architecture over the other. We fully agree that the reader should get a full picture and, hence, in light of reviewer's comment, we have added a paragraph to address this comparison between our proposed architecture and the PDH loop (page 2, line 14).

“Using the PDH technique, a sharp asymmetric error signal can be generated, which can then be utilized as a servo signal to stabilize the laser frequency. In PDH architecture, the laser phase is modulated using a phase modulator, filtered by an optical frequency reference (e.g. Fabry-Perot (FP) cavity) followed by a photodetector [28]. The photodetected signal is down-converted with the same local oscillator frequency to base-band. This up/down conversion scheme makes PDH more robust against DC error signal offset, base-band noise, and other perturbations at low frequency which enables a robust stable frequency lock.”

Regarding the reviewer's comment about the effect of laser intensity noise, we have added a section in Supplementary Note 4 (page 7, Supplementary Information). In our proposed architecture, the laser intensity noise is a common noise detected by two nominally identical integrated photodetectors. Due to the balanced photodetector arrangement in our proposed architecture, the laser intensity noise will be significantly suppressed by the common-mode-rejection ratio of the balanced photodetectors.

In light of reviewer's comment, we have added Supplementary Note 5 to discuss this analytically and also have added a sentence on page 3, line 12:

“Moreover, the proposed architecture utilizes a balanced detection scheme which significantly suppresses the intensity noise of the laser by the common mode rejection ratio of the balanced photodetectors [32].”

Another point: of the applications mentioned in the introduction, it is difficult to believe that this specific demonstration - at least in its current form - will have an impact on optical atomic clocks as the laser linewidths required for a clock interrogation laser are at between 4 and 6 orders of magnitude better than demonstrated here.

Thank you for your comment. We agree with the reviewer that for some applications like optical atomic clocks, lower level of laser frequency noise is required, however, as we mentioned earlier in this document, we would like to strongly emphasize that we are not proposing a new laser technology; we are proposing a new laser frequency locking technique that, similar to PDH or other techniques, can be applied to any laser and improve integrated and bench-top state-of-the-art laser locking systems. Our proposed laser frequency locking method is simpler, effective and promising to be implemented at scale which, at the end, would benefit those applications mentioned in the introduction.

We are demonstrating this technique in silicon chip as a proof-of-concept with all the benefits and short-comings discussed in the manuscript (e.g. page 5, paragraph 2). We are aware of limitations of this material platform and have addressed them comprehensively through measurement, analysis, simulation, and literature review. As we have discussed in the Discussion section, the

choice of material platform and cavity is up to the designer and, indeed, is application specific. In other words, one may decide to pick a inexpensive laser with MHz linewidth and use this architecture in a low-cost, mature, scalable, and small footprint silicon chip but in another application (e.g. atomic clocks) the same proposed architecture can be applied to a better laser used with a better frequency reference to lock free-running narrow linewidth lasers with kHz linewidth to achieve Hz-level final spectral linewidth. We believe our revised manuscript along with the modified Supplementary Information document assist readers in replicating our work, understanding the trade-offs, and adapting our proposed architecture for their specific needs.

We have added the following sentence to make our point clear (page 3, paragraph 3):

“...We emphasize that our proposed laser stabilization technique is versatile and, similar to other laser frequency locking techniques (e.g., PDH), it can be applied across various platforms with a properly chosen frequency reference to achieve laser frequency noise suppression.”

At the end of the manuscript the authors provide routes for reducing the TRN such as using a SiN ring or ultralow loss Si. This is important; however, this comparison requires some numbers. What kind of improvement in frequency noise or linewidth is expected? At what volume/area cost? Similarly, for a macroscopic cavity: the authors could give one example of the noise improvement and corresponding system volume increase if this method were implemented on a standard stable reference cavity. I don't believe that simply saying that they would provide better performance at the cost of higher chip area provides the reader enough information.

Thank you for your comments and feedback. In light of the reviewer's comments and suggestions, we have added Supplementary Note 4 (page 10, Supplementary Information) to numerically discuss calculation of the microresonator TRN. Moreover, we have discussed different optical frequency reference technologies and the trade-off between cavity volume and achievable TRN on page 12, 13 (Discussion section) and Fig. 6 of the main manuscript.

Fig. 6. Thermorefractive noise in optical cavities. Cavity TRN limit for different optical cavities at 1 kHz offset frequency shows a clear trend where TRN reduces as the cavity mode volume increases.

Moreover, Supplementary Note 4 discusses the closed-loop operation in presence of noise sources and includes an example demonstrating the frequency noise PSD and linewidth estimation of a laser locked to a high-Q silicon nitride microresonator with significantly lower TRN.

Supplementary Figure 8 | Simulation of the frequency noise PSD of a laser locked to a SiN microring. **a** The normalized ring response (blue) of the silicon nitride (SiN) microresonator. The error signal (red), I_{error} , is asymmetric around the ring resonance. **b** The free-running and stabilized laser frequency noise PSD.

Finally, Supplementary Note 6 discusses system level design procedures and considerations. In addition to the Discussion section, we have added the following sentence on page 12, line 5:

“Supplementary Note 4 discusses the closed-loop operation in presence of noise sources and includes an example demonstrating the frequency noise PSD and linewidth estimation of a laser locked to a high-Q silicon nitride microresonator with significantly lower TRN.”

- Does the work support the conclusions and claims, or is additional evidence needed?

Mostly. A few comments though:

One interesting question is the “line-splitting” factor achieved in this demonstration. The resonance width is 80 MHz and final linewidth 700 kHz, implying a line-splitting of $\sim 100\times$. However, if the limitation is the TRN of the ring, as the authors state, this is a limitation of the device, not a limitation of the method. It would be interesting if the authors showed an in-loop error signal PSD measurement to show that the method would be able to support references with lower thermal noise. Essentially, I am asking for a measurement of the residual noise of the lock $(S_0(f)/(KE*KL*KFD)^2 + S_n(f)/KFD^2)$, as this could show the power of this method if it were applied to a different, better resonator and would strengthen the manuscript.

We appreciate your comment and suggestions. We certainly think this is an interesting measurement and would strengthen our work. In light of the reviewer’s suggestion, we have measured and reported the in-loop PSD of the error signal (please see the image below), as shown in figure 5. As shown in Fig. 5(a), the error signal in voltage domain is sampled using a low

noise FPGA (Redpitaya STEMLab 125-14) and used for in-loop frequency noise PSD processing. We have measured the frequency noise of three different DFB lasers and their corresponding in-loop frequency noise PSD as shown in Fig. 5(b-d). As suggested by Fig. 5(b-d), the in-loop frequency noise PSD goes down to $< 10^2 \text{ Hz}^2/\text{Hz}$ which is more than 3 orders-of-magnitude less than the silicon microresonator TRN at 1 kHz Fourier frequency. This clearly shows that the implemented laser frequency stabilization method can suppress the laser frequency noise significantly and in the current demonstration is limited by the silicon microresonator's TRN. In our new experiment, given the loop bandwidth, we observed about 40 dB frequency noise suppression at 1.5 kHz offset frequency. Moreover, the rms frequency noise is suppressed from 3.93 MHz to 398 kHz for DFB1, 3 MHz to 207 kHz for DFB2, and 1.37 MHz to 136 kHz for DFB3 within the loop bandwidth. In our experiments, the integral linewidth of DFB3 is reduced from 3.23 MHz to 321 kHz, due to lower free-running frequency noise and higher servo loop bandwidth.

Fig. 5. The closed-loop operation. a Measurement setup for laser frequency locking experiment. Part of the laser output is beat with the output of the fully-stabilized optical frequency comb and photodetected using a fast receiver. The beat note is then digitized and used for frequency noise analysis. Heaters are off during this measurement. b-d The power spectral density of frequency noise of three different DFB lasers under free-running and closed-loop operation. The in-loop error signal is digitized and processed using an FPGA for in-loop frequency noise power spectral density (PSD) estimation. The frequency noise of the stabilized DFB lasers are limited to thermorefractive noise (TRN) of the silicon microring. The RBW of laser frequency noise measurements is 200 Hz. RBW of in-loop noise measurements in (b) and (c) is 51 Hz, and in (d) is 103 Hz, respectively. (b-d) share the same vertical axis. EDFA: Erbium-doped fiber amplifier, FP: Fabry-Perot, FPGA: field programmable gate array.

We have changed the closed-loop experiment section on pages 9-12 accordingly and added the following paragraph in page 11, paragraph 2 to address in-loop noise measurement:

“The in-loop error signal PSD (Fig. 5a) reveals the relative frequency noise PSD between the laser and the microresonator, indicating negligible impact from absolute cavity TRN. Therefore, it can be used to estimate the frequency locking performance and ultimate frequency noise suppression, assuming same loop parameter (e.g. gain and noise) but an ideal cavity with negligible TRN. As shown in Fig. 5b-d, the in-loop frequency noise PSD suggests that, indeed, the dominant limiting noise contribution in the loop is the microring TRN, and the proposed frequency stabilization loop is capable of suppressing close-in frequency noise to less than $10^2 \text{ Hz}^2 \cdot \text{Hz}^{-1}$ (extra 30 dB frequency noise suppression), emphasizing the effectiveness of the proposed approach.”

The authors did the initial characterization of their ring with an ECDL and then they switched to a DFB laser for the lock demonstration. The reason for this change is not stated though I suspect this is because the linewidth of the ECDL was too narrow to show an improvement in the demonstration experiment. If this is the case, I understand that the demonstration requires a laser that is bad enough such that it can be improved with the method, but this is a point that needs to be explicitly acknowledged so the reader does not have to guess.

Thank you for your comment and valuable feedback. The purpose of using an ECDL with lower frequency noise, a linear frequency chirp, and a graphical user interface was to facilitate accurate characterization and report the silicon ring resonator and the error signal responses (Fig. 4(b, c)). We didn't intend to use that laser for our closed-loop experiment. As the reviewer mentioned correctly, given the TRN of our silicon cavity we needed a DFB laser with noise level high enough so we can show reasonable frequency noise suppression. We have added a sentence to make this point clear to the readers (page 8, line 15).

“The narrow linewidth ECDL with linear frequency chirp facilitates accurate characterization and calibration of the ring resonator and the error signal responses.”

And also (page 9, line 11):

“As a proof-of-concept demonstration of laser frequency locking, given the TRN of the integrated micro-ring resonator, three different commercially available DFB lasers at wavelength of 1550.7 nm, 1547.8 nm, and 1551.4 nm with large enough frequency noise are chosen. Moreover, the electronic noise of the lasers bias current is adjusted and controlled to accentuate laser frequency locking demonstration.”

- Are there any flaws in the data analysis, interpretation and conclusions? - Do these prohibit publication or require revision?

Not a major flaw, but this manuscript would benefit a few-sentence and few-equation supplement showing the TRN calculation. This way the reader can double-check and/or use the same methodology to calculate the TRN for a different ring.

Thank you for your comments and suggestions. In light of the reviewer's comments, we have added a Supplementary Note 4 to discuss TRN calculation of a cavity (page 11, Supplementary Information).

Supplementary Figure 6 | Thermorefractive noise of a silicon microring resonator. The TRN noise of microring resonator described in the main manuscript is simulated with COMSOL FEM simulation (green) and compared with analytical model (blue).

- Is the methodology sound? Does the work meet the expected standards in your field?

Generally, yes.

We appreciate your comment and valuable feedback.

- Is there enough detail provided in the methods for the work to be reproduced?

The method itself can likely be reproduced in another experiment with the information available. I do not have enough expertise on device fabrication to say whether there is enough information to make the same device.

Thank you for your comment. We believe a reader should be able to reproduce our proposed architecture in another experiment given the detailed information provided in the manuscript and the Supplementary Information document.

As far as the design of the SOI chip is concerned, we have included details of the silicon cavity design (including dimensions) in Fig. 3. We have also mentioned extra details in the Methods section ("Photonic chip implementation") plus Supplementary Note 6 to add even more details on design procedure and considerations.

For device fabrication, we utilize the commercially available AIM Photonics 180 nm SOI process, as mentioned in the abstract, introduction (page 3 paragraph 2), and the Methods section. This is a mature and reliable silicon photonic foundry with publicly available dedicated and multi-project wafer (MPW) services.

-Some typos:

In the first introduction paragraph:

“The Optical feedback method relies “  Perhaps there is no need to capitalize optical

In the last sentence of section I:

“The proposed architecture offers a promising solution for achieving sensitive, simple, and low power”  simple

In section II(D):

“A tunable continuous wave laser (TOPTICA CTL 1550)”  continuous

-Other minor comments:

“ $|T(\omega)|$, $\psi(\omega)$, ϕ , P_0 , and R are the amplitude and phase of the optical reference at the frequency of ω ”  Amplitude and phase of the reflection coefficient of the optical reference.

Thank you for your comments. We have fixed these typos in the main manuscript.

List of major modifications/additions in the revised manuscript

- **Main manuscript:**

- 1) New measurement setup: we have changed the frequency noise measurement setup from fiber based delayed-self heterodyne to heterodyne beat note measurement using fully stabilized optical frequency comb as shown in Fig. 5(a). We have also used an FPGA for in-loop noise analysis.
- 2) New measurement: the closed-loop laser frequency locking experiment is repeated for three different DFB lasers and reported in Fig. 5(b-d).
- 3) New measurement: The PSD of the in-loop error signal is measured and reported in Fig. 5(b-d) for three DFB laser frequency locking experiment.
- 4) New figure: in light of reviewers comments, we have added Fig. 6 to compare different cavity technologies in terms of volume and achieved TRN to give readers a more complete picture.
- 5) New Method: we have added a new paragraph in the Method section to explain details of the new heterodyne beat note measurement setup.
- 6) New references: New references have been added to the reference list.
- 7) New text: in light of reviewers comments and according to new major modifications in our paper, we have modified and added text in the main manuscript which has already discussed earlier in the point-by-point response to reviewers comments.

- **Supplementary Information:**

- 1) New Supplementary Note: we have added Supplementary Note 4, discussing noise sources in the loop using analytical modeling and numerical simulations.
- 2) New Supplementary Note: we have added Supplementary Note 6, to discuss design procedure and considerations.
- 3) Modified Supplementary Note: In light of the reviewer's comment, a numerical simulation of laser frequency noise PSD and linewidth estimation is included.
- 4) New Supplementary Note: a new Supplementary Table 1 is added that includes model parameters for numerical simulation in Supplementary Note 5.
- 5) New Supplementary Table: according to our new measurement in the revise manuscript, the Supplementary Table 2 (list of all equipment and tools) has been updated.
- 6) New text: the order of the notes in the Supplementary Information has been changed accordingly. New text has been added to the Supplementary Information accordingly.

REVIEWERS' COMMENTS

Reviewer #1 (Remarks to the Author):

I appreciate the authors for well addressing all comments, and I strongly support the publication of the manuscript in Nature Communications. However, I have one comment left:

- Why is the frequency noise shown only at offset frequencies above 1 kHz? As I understand that is because at lower offset frequencies the noise is not limited by the TRN. However, the frequency noise at lower offset frequencies should be shown at least in Supplementary Material.

Typos:

- Page 3 line 17: photo detector
- Page 9 line 6: oscilloscope .

Reviewer #2 (Remarks to the Author):

The authors have addressed my questions and concerns.

Response to reviewers comments

First and foremost, we would like to thank all the reviewers for their time and valuable comments that have certainly improved our manuscript. Please find our point by point response to all comments below. Our responses are shown in blue and the changes made in the revised manuscript are shown in red.

Reviewer #1 (Remarks to the Author):

I appreciate the authors for well addressing all comments, and I strongly support the publication of the manuscript in Nature Communications.

We sincerely appreciate the valuable comments provided by the reviewer, which have strengthened our manuscript.

However, I have one comment left:

- Why is the frequency noise shown only at offset frequencies above 1 kHz? As I understand that is because at lower offset frequencies the noise is not limited by the TRN. However, the frequency noise at lower offset frequencies should be shown at least in Supplementary Material.

The close-in frequency noise (e.g. < 1 kHz) is, indeed, important for low noise lasers that are locked to a stable cavity with excellent long-term stability. In such cases, when the β -separation line intersects with the frequency noise PSD at lower frequencies (low integration BW), it is crucial to extend the noise measurement to lower frequency offsets to have an accurate estimation of the integral linewidth. In our case, however, the β -separation line intersection with the frequency noise PSD of the three locked DFB lasers are between 100 kHz - 1 MHz (Fig. 5b-d). So, we have already captured enough spectrum to be able to accurately estimate the integral linewidth suppression factor for our proof-of-concept demonstration.

Accurate measurement at much lower offset frequencies necessitates longer beat-note recording times and modifications to the chip packaging and the printed circuit board for improved thermal insulation and temperature stabilization, which are challenging in the current setup but certainly can be improved in the future versions.

With that said, in response to the reviewer's comment, we have modified Fig. 5 to illustrate the frequency noise PSDs for offsets below 1 kHz (down to 500 Hz) to show the PSD trend more clearly. We believe this adjustment provides a clear enough representation of the PSD and laser noise behavior for the proof-of-concept demonstration.

The integral linewidth calculation and rms frequency noise estimations are updated accordingly (page 11):

“As can be seen from Fig. 5b, the root mean square (rms) of laser frequency noise ($\sigma_{\delta f}$) is suppressed from 4.28 MHz to 400.3 kHz within the loop bandwidth of 80 kHz. Figure 5c shows the measured PSD of the frequency noise of DFB laser two, lasing at wavelength of 1547.8 nm. As shown in Fig. 5c, the free-running laser frequency noise is suppressed by more than 42 dB at about 1.5 kHz Fourier frequency. Within the loop bandwidth of 100 kHz, $\sigma_{\delta f}$ of the free-running DFB laser two is reduced from approximately 3.87 MHz to about 240.3 kHz. Figure 5d shows the measured PSD of free-running and stabilized DFB laser three at wavelength of 1551.4 nm. As shown in Fig. 5d, the frequency noise is suppressed by 39 dB at 1.5 kHz Fourier frequency and $\sigma_{\delta f}$ of laser frequency noise within the loop bandwidth of 150 kHz is suppressed from 1.7 MHz to 140.1 kHz.”

(page 11, paragraph 3):

“While the frequency locked DFB1 and DFB2 demonstrate a significant reduction in close-in frequency noise, their integral linewidth is suppressed by only a factor of 2.1 and 2.9, respectively, attributed to a larger initial linewidth and a smaller servo bump bandwidth. In contrast, the integral linewidth of stabilized DFB3 drops from 4 MHz to 330 kHz (approximately a factor of 12), thanks to a higher servo bump frequency and a lower free-running Lorentzian linewidth (Fig. 5d).”

Typos:

- Page 3 line 17: photo detector
- Page 9 line 6: oscilloscope .

Thank you, the typos are fixed.

Reviewer #2 (Remarks to the Author):

The authors have addressed my questions and concerns.

We sincerely appreciate the valuable comments provided by the reviewer, which have strengthened our manuscript.